# GDP: Generalized Device Placement for Dataflow Graphs

## Abstract

Runtime and scalability of large neural networks can be significantly affected by the placement of operations in their dataflow graphs on suitable devices. With increasingly complex neural network architectures and heterogeneous device characteristics, finding a reasonable placement is extremely challenging even for domain experts. Most existing automated device placement approaches are impractical due to the significant amount of compute required and their inability to generalize to new, previously held-out graphs. To address both limitations, we propose an efficient end-to-end method based on a scalable sequential attention mechanism over a graph neural network that is transferable to new graphs. On a diverse set of representative deep learning models, including Inception-v3, AmoebaNet, Transformer-XL, and WaveNet, our method on average achieves 16% improvement over human experts and 9.2% improvement over the prior art with $15\times$ faster convergence. To further reduce the computation cost, we pre-train the policy network on a set of dataflow graphs and use a superposition network to fine-tune it on each individual graph, achieving state-of-the-art performance on large hold-out graphs with over 50k nodes, such as an 8-layer GNMT.

## 1 Introduction

Neural networks have demonstrated remarkable scalability–improved performance can usually be achieved by training a larger model on a larger dataset (Hestness et al., 2017; Shazeer et al., 2017; Jozefowicz et al., 2016; Mahajan et al., 2018; Radford et al.). Training such large models efficiently while meeting device constraints, like memory limitations, necessitate partitioning of the underlying dataflow graphs for the models across multiple devices. However, devising a good partitioning and placement of the dataflow graphs requires deep understanding of the model architecture, optimizations performed by domain-specific compilers, as well as the device characteristics, and is therefore extremely hard even for experts.

ML practitioners often rely on their understanding of model architecture to determine a reasonable partitioning and placement for graphs. However, relying solely on the model architecture while ignoring the effect of the partitioning on subsequent compiler optimizations like op-fusion can lead to sub-optimal placements and consequently under-utilization of available devices. The goal of automated device placement is to find the optimal assignment of operations to devices such that the end-to-end execution time for a single step is minimized and all device constraints like memory limitations are satisfied. Since this objective function is non-differentiable, prior approaches (Mirhoseini et al., 2017; 2018; Gao et al., 2018) have explored solutions based on reinforcement learning (RL). However, these RL policies are usually not transferable and require training a new policy from scratch for each individual graph. This makes such approaches impractical due to the significant amount of compute required for the policy search itself, at times offsetting gains made by the reduced step time.

In this paper, we propose an end-to-end deep RL method for device placement where the learned policy is generalizable to new graphs. Specifically, the policy network consists of a graph-embedding network that encodes operation features and dependencies into a trainable graph representation, followed by a scalable sequence-to-sequence placement network based on an improved Transformer (Vaswani et al., 2017; Dai et al., 2019). The placement network transforms the graph representations into a placement decision with soft attention, removing hard constraints such as hierarchical

grouping of operations (Mirhoseini et al., 2018) or co-location heuristics (to reduce the placement complexity) (Mirhoseini et al., 2017). Both of our graph-embedding network and placement network can be jointly trained in an end-to-end fashion using a supervised reward, without the need to manipulate the loss functions at multiple levels. We empirically show that the network learns flexible placement policies at a per-node granularity and can scale to problems over 50,000 nodes.

To generalize to arbitrary and held-out graphs, our policy is trained jointly over a set of dataflow graphs (instead of one at a time) and then fine-tuned on each graph individually. By transferring the learned graph embeddings and placement policies, we are able to achieve faster convergence and thus use less resources to obtain high-quality placements. We also use super-positioning, i.e., a feature conditioning mechanism based on the input graph embeddings, to effectively orchestrate the optimization dynamics of graphs with drastically different sizes in the same batch.

Our contributions can be summarized as follows:

1. An end-to-end device placement network that can generalize to arbitrary and held-out graphs. This is enabled by jointly learning a transferable graph neural network along with the placement network.

2. A scalable placement network with an efficient recurrent attention mechanism, which eliminates the need for an explicit grouping stage before placement. The proposed end-to-end network provides $15\times$ faster convergence as compared to the hierarchical LSTM model used in earlier works (Mirhoseini et al., 2017; 2018).

3. A new batch pre-training and fine-tuning strategy based on network superposition, which leads to improved transferability, better placements especially for larger graphs, and $10\times$ reduction in policy search time as compared to training individual graphs from scratch.

4. Superior performance over a wide set of workloads, including InceptionV3 (Szegedy et al., 2015), AmoebaNet (Real et al., 2018), RNNs, GNMT (Wu et al., 2016), Transformer-XL (Dai et al., 2019), WaveNet (van den Oord et al., 2016), and more.

## 2 RELATED WORK

**Device Placement**   Reinforcement learning has been used for device placement of a given dataflow graph (Mirhoseini et al., 2017) and demonstrated run time reduction over human crafted placement and conventional heuristics. For improved scalability, a hierarchical device placement strategy (HDP) (Mirhoseini et al., 2018) has been proposed that clusters operations into groups before placing the operation groups onto devices. Spotlight (Gao et al., 2018) applies proximal policy optimization and cross-entropy minimization to lower training overhead. Both HDP and Spotlight rely on LSTM controllers that are difficult to train and struggle to capture very long-term dependencies over large graphs. In addition, both methods are restricted to process only a single graph at a time, and cannot generalize to arbitrary and held-out graphs. Placeto (Addanki et al., 2019) represents the first attempt to generalize device placement using a graph embedding network. But like HDP, Placeto also relies on hierarchical grouping and only generates placement for one node at each time step. Our approach (GDP) leverages a recurrent attention mechanism and generates the whole graph placement at once. This significantly reduces the training time for the controller. We also demonstrate the generalization ability of GDP over a wider set of important workloads.

**Parallelization Strategy**   Mesh-TensorFlow is a language that provides a general class of distributed tensor computations. While data-parallelism can be viewed as splitting tensors and operations along the "batch" dimension, in Mesh-TensorFlow the user can specify any tensor-dimensions to be split across any dimensions of a multi-dimensional mesh of processors. FlexFlow (Jia et al., 2018) introduces SOAP, a more comprehensive search space of parallelization strategies for DNNs which allows parallelization of a DNN in the Sample, Operator, Attribute, and Parameter dimensions. It uses guided randomized search of the SOAP space to find a parallelization strategy for a specific parallel machine. GPipe (Huang et al., 2018) proposed pipeline parallelism, by partitioning a model across different accelerators and automatically splitting a mini-batch of training examples into smaller micro-batches. By pipelining the execution across micro-batches, accelerators can operate in parallel. Our GDP focuses on a general deep RL method for automating device placement on arbitrary graphs, and is therefore orthogonal to existing parallelization strategies.

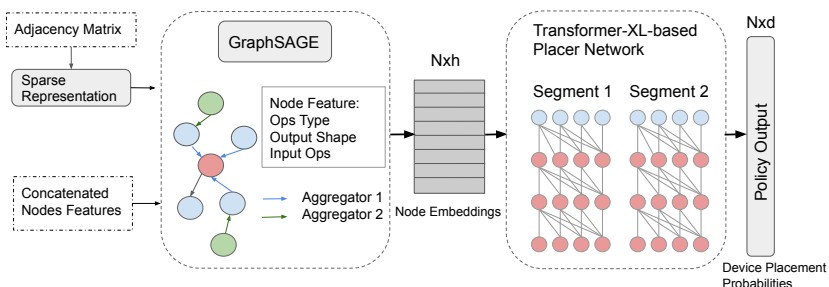

Figure 1: Overview of GDP: An end-to-end placement network that combines graph embedding and sequential attention. $N$: Number of Nodes, $h$: Hidden Size, $d$: Number of Devices.

**Compiler Optimization** REGAL (Paliwal et al., 2019) uses deep RL to optimize the execution cost of computation graphs in a static compiler. The method leverages the policy's ability to transfer to new graphs to improve the quality of the genetic algorithm for the same objective budget. However, REGAL only targets peak memory minimization while GDP focuses on graph run time and scalability while also meeting the peak memory constraints of the devices. Specifically, we generalize graph partitioning and placement into a single end-to-end problem, with and without simulation, which can handle graphs with over 50,000 nodes.

## 3 END-TO-END PLACEMENT POLICY

Given a dataflow graph $G(V, E)$ where $V$ represents atomic computational operations (ops) and $E$ represents the data dependency, our goal is to learn a policy $\pi : \mathcal{G} \mapsto \mathcal{D}$ that assigns a placement $D \in \mathcal{D}$ for all the ops in the given graph $G \in \mathcal{G}$, to maximize the reward $r_{G,D}$ defined based on the run time. $\mathcal{D}$ is the allocated devices that can be a mixture of CPUs and GPUs. In this work, we represent policy $\pi_\theta$ as a neural network parameterized by $\theta$.

Unlike prior works that focus on a single graph only, the RL objective in GDP is defined to simultaneously reduce the expected runtime of the placements over a set of $N$ dataflow graphs:

$$J(\theta) = \mathbb{E}_{G \sim \mathcal{G}, D \sim \pi_\theta(G)}[r_{G,D}] \approx \frac{1}{N} \sum_G \mathbb{E}_{D \sim \pi_\theta(G)}[r_{G,D}] \tag{1}$$

In the following, we refer to the case when $N = 1$ as individual training and the case when $N > 1$ as *batch training*. We optimize the objective above using Proximal Policy Optimization (PPO) (Schulman et al., 2017) for improved sample efficiency.

Figure 1 shows an overview of the proposed end-to-end device placement network. Our proposed policy network $\pi_\theta$ consists a graph embedding network that learns the graphical representation of any dataflow graph, and a placement network that learns a placement strategy over the given graph embeddings. The two components are jointly trained in an end-to-end fashion. The policy $p(a|G)$ is applied to make a set of decisions at each node. These decisions, denoted as $a_v$ for each $v \in V$ across all nodes, form one action $a = \{a_{v \in V}\}$. One decision corresponds to playing one arm of a multi-bandit problem, and specifying the entire $a$ corresponds to playing several arms together in a single shot. Note the architecture is designed to be invariant over the underlying graph topology, enabling us to apply the same learned policy to a wide set of input graphs with different structures.

### 3.1 GRAPH EMBEDDING NETWORK

We leverage graph neural networks (GNNs) (Hamilton et al., 2017; Xu et al., 2019; You et al., 2018) to capture the topological information encoded in the dataflow graph. Most graph embedding frameworks are inherently transductive and can only generate embeddings for a given fixed graph. These transductive methods do not efficiently extrapolate to handle unseen nodes (e.g., in evolving graphs), and cannot learn to generalize to unseen graphs. GraphSAGE (Hamilton et al., 2017) is an inductive framework that leverages node attribute information to efficiently generate representations

on previously unseen data. While our proposed framework is generic, we adopt the feature aggregation scheme proposed in GraphSAGE to model the dependencies between the operations and build a general, end-to-end device placement method for a wide set of dataflow graphs.

In GDP, nodes and edges in the dataflow graph are represented as the concatenation of their meta features (e.g., operation type, output shape, adjacent node ids) and are further encoded by the graph embedding network into a trainable representation. The graph embedding process consists of multiple iterations, and the computation procedure for the $l$-th iteration can be outlined as follows:

First, each node $v \in V$ aggregates the feature representations of its neighbors, $\{h_u^{(l)}, \forall u \in \mathcal{N}(v)\}$, into a single vector $h_{\mathcal{N}(v)}^{(l)}$. This aggregation outcome is a function of all previously generated representations, including the initial representations defined based on the input node features. In this work, we use the following aggregation function with max pooling:

$$h_{\mathcal{N}(v)}^{(l)} = \max(\sigma(W^{(l)} h_u^{(l)} + b^{(l)}), \forall u \in \mathcal{N}(v)) \tag{2}$$

where $(W^{(l)}, b^{(l)})$ define an affine transform and $\sigma$ stands for the sigmoid activation function. We then concatenate the node's current representation, $h_v^{(l)}$, with the aggregated neighborhood vector, $h_{\mathcal{N}(v)}^{(l)}$, and feed this concatenated vector through a fully connected layer $f^{(l+1)}$

$$h_v^{(l+1)} = f^{(l+1)}(\text{concat}(h_v^{(l)}, h_{\mathcal{N}(v)}^{(l)})) \tag{3}$$

Different from GraphSAGE, parameters in our graph embedding network are trained jointly with a placement network via stochastic gradient descent with PPO, in a *supervised* fashion, as described in Section 3. That is, we replace the unsupervised loss with our task-specific objective.

## 3.2 PLACEMENT NETWORK

The graph neural network works as a feature aggregation network that learns a trainable feature representation for the computational graph, we still need a policy network that produces actions on a per node basis. Given $h_v$'s, the policy network produces $a_v$'s through conditionally independent predictions, where the prediction for one node $v$ does not depend on the prediction of other nodes.

$$p(a|G) = \prod_v p(a_v|G) = \prod_v p(a_v|f(h_v)) \tag{4}$$

While $f$ can be represented using multilayer perceptrons (MLPs), where the MLPs is shared across all nodes for prediction the placement output distributions. However, MPLs lack a dependency tracking mechanism across nodes. In practise, the placement of one node can be determined by the placement of another node, where the placed node may consume a large size of data produced by the other node. Intuitively, an attention network can learn this dependency and the relative importance of dependencies across an entire graph. Therefore, we decide to use an attention-based placement network to better track inter-node placement-related dependencies.

Designing a scalable placement network that can generalize to graphs with thousands of nodes is challenging, as the conventional GNMT models proposed for language tasks usually target a shorter sequence length. Hierarchical placement (Mirhoseini et al., 2018) has been proposed to address this issue,however, the proposed grouper network comes with limited flexibility and generality. For example, the grouper network leverages an aggregated feature representation by averaging feature vectors for nodes within the same group. The non-differentiable grouping procedure prevents training the graph-embedding and placement networks end-to-end.

To remove the two-stage hierarchical workflow in HDP for improved scalability, we propose to use a Transformer-based attentive network to generate operation placements in an end-to-end fashion. As the graph embedding already contains spatial (topological) information for each node, we remove the positional embedding in the original transformer to prevent the model from overfitting node identifications. To capture long-term dependencies efficiently among a large set of nodes, we adopt segment-level recurrence introduced in Transformer-XL (Dai et al., 2019; Dai, 2019), where hidden states computed for the previous set of nodes are cached (with gradient flows disabled) and reused as an extended context during the training of the next segment. Besides achieving extra long context,

we empirically find the segment-level recurrent attention much faster than a conventional LSTM-based GNMT model. In our experimental evaluation, we compare both the performance and speed up of our placement network with that of the LSTM-based hierarchical device placement.

## 3.3 BATCH TRAINING WITH PARAMETER SUPERPOSITION

Since the parameterization for the architecture of the end-to-end policy is designed to be invariant over input graphs with different topologies, the same placement policy can be shared across a wide set of workloads. We therefore propose a batch training strategy, and further enhance the aforementioned architecture to handle such generalization across graphs.

Naïve batch training is challenging in our context as different dataflow graphs contain different number of operations connected in different topologies. In addition, unlike previous device placement methods, GDP aims to handle graphs from potentially different application domains (e.g. computer vision, language, and speech), where the number of operations can range from a few thousand to one million. These graphs have drastically different network architecture, in terms of computational operations, data shape, and network topology. As an example, recurrent networks have completely different operation types and connections compared to multi-branch convolutional networks that are widely used in computer vision. It would be highly desirable to train a single shared network that maximizes information sharing across these heterogeneous tasks, without hurting the performance on each of them due to their distinct learning dynamics.

Along a similar direction of multi-task learning and few-shot learning (Oreshkin et al., 2018), we propose a feature conditioning mechanism similar to *parameter superposition* (Cheung et al., 2019). The idea is to train one shared policy, but condition its parameters based on the input features to mitigate the potentially undesirable interference among different input graphs. Since dense layers (affine transforms followed by nonlinearity) serve as the fundamental building blocks in all of our network components, we introduce an additional conditioning layer to enable superposition in all dense layers the placement network:

$$x^{(l+1)} = g^{(l)}(c(x^{(0)}) \odot x^{(l)}) \tag{5}$$

where $g^{(l)}$ stands for a dense layer in our policy network, $c$ stands for the feature conditioning layer, and $x^{(0)}$ denotes the feature representation of the input graph generated by the graph-embedding network. The feature conditioning layer is implemented with minimum overhead by adding an additional transformer layer to our placement network.

## 4 EXPERIMENT

### 4.1 EXPERIMENT SETUP

In this section, we evaluate our training strategy on widely used machine learning models in computer vision, natural language processing, and speech domains. We compare our approach to human expert placement, TensorFlow METIS placement, and hierarchical device placement (HDP) (Mirhoseini et al., 2018). Our experiments are run on machines with one Intel Broadwell CPU and up to eight Nvidia P100 GPUs. Note that the prior work (Mirhoseini et al., 2017; 2018; Gao et al., 2018) were evaluated on different GPU devices, preventing direct comparison of results. Therefore, we re-evaluate HDP on our own system environment and report those numbers.

The performance of a placement is evaluated by the resulted training step time (run time) of the neural network. We use the negative square root of the normalized run time as the reward, where the run time is normalized with the best run time from a baseline. We use the average reward of all the previous trials as a bias term. The advantage value is computed by subtracting the reward by the average reward. During the search, we apply a large negative reward (-10) for invalid placements (e.g. a violation of co-location constraint, out of memory, etc.). For operation scheduling, we rely on the Tensorflow default FIFO scheduling.

### 4.2 PERFORMANCE ON INDIVIDUAL GRAPHS

We evaluate GDP by training the model separately on six important graphs, including RNN Language Modeling, GNMT (Sutskever et al., 2014), Transformer-XL, Inception, AmoebaNet, and

Table 1: Run time comparison between GDP-one, human expert, Tensorflow METIS, and hierarchical device placement (HDP) on six graphs (RNNLM, GNMT, Transformer-XL, Inception, AmoebaNet, and WaveNet). Graph runtime speed up is compared with Human Placement (HP) and Hierarchical Device Placement (HDP). Search speed up is the policy network training time speed up compared to HDP (reported values are averages of six runs).

| Model (#devices) | GDP-one (s) | HP (s) | METIS (s) | HDP (s) | Run time speed up over HP / HDP | Search speed up |
|---|---|---|---|---|---|---|
| 2-layer RNNLM (2) | 0.234 | 0.257 | 0.355 | 0.243 | 9.8% / 4% | 2.95x |
| 4-layer RNNLM (4) | 0.409 | 0.48 | OOM | 0.490 | 17.4% / 19.8% | 1.76x |
| 2-layer GNMT (2) | 0.301 | 0.384 | OOM | 0.376 | 27.6% / 24.9% | 30x |
| 4-layer GNMT (4) | 0.409 | 0.469 | OOM | 0.520 | 14.7% / 27.1% | 58.8x |
| 8-layer GNMT (8) | 0.649 | 0.610 | OOM | 0.693 | -6% / 6.8% | 7.35x |
| 2-layer Transformer-XL (2) | 0.386 | 0.473 | OOM | 0.435 | 22.5% / 12.7% | 40x |
| 4-layer Transformer-XL (4) | 0.580 | 0.641 | OOM | 0.621 | 11.4% / 7.1% | 26.7x |
| 8-layer Transformer-XL (8) | 0.748 | 0.813 | OOM | 0.789 | 8.9% / 5.5% | 16.7x |
| Inception (2) | 0.405 | 0.418 | 0.423 | 0.417 | 3.2% / 3% | 13.5x |
| AmoebaNet (4) | 0.394 | 0.44 | 0.426 | 0.418 | 26.1% / 6.1% | 58.8x |
| 2-stack 18-layer WaveNet (2) | 0.317 | 0.376 | OOM | 0.354 | 18.6% / 11.7% | 6.67x |
| 4-stack 36-layer WaveNet (4) | 0.659 | 0.988 | OOM | 0.721 | 50% / 9.4% | 20x |
| GEOMEAN | - | - | - | - | **16% / 9.2%** | **15x** |

WaveNet. We name this approach **GDP-one**. For all the tasks, GDP-one consistently outperforms human expert placement, TensorFlow METIS (Karypis & Kumar, 1998) placement, and HDP. For extremely large graphs, GDP-one is only 6% worse on 8-layer NMT (over 60k nodes), compared to human placement, but is 6.8% better than HDP. Overall, GDP-one achieves on average more than 16% run time reduction across the evaluated 12 graphs, compared to human expert placement. Compared to hierarchical device placement, GDP-one achieves an average 9.2% speed up, and scales better to large graphs such as 8-layer NMT and 4-layer RNNLM. Importantly, with the efficient end-to-end training and sample efficient reinforcement learning algorithm, GDP-one has a 15x speed up in convergence time of the placement network over HDP.

### 4.3 GENERALIZATION

GDP enables the training of multiple heterogeneous graphs in a single batch, sharing parameters in the graph-embedding network and the placement network. We name this training strategy **GDP-batch**. We empirically show that GDP-batch generates better placements for many workloads such as transformer-XL (7.6%), WaveNet (15%), and 8-layer GNMT (8%). Table 2 compares the run time of 11 tasks using GDP-batch, with the same end-to-end architecture as described in section 4.2. GDP-batch yields slightly better run time compared to GDP-one in majority of the tasks, while being only slightly worse on AmoebaNet. Compared to training graphs separately, GDP-batch reduces network parameters and enables transfer learning among different graphs.

We further evaluate the effect of transfer learning by mixing redundant tasks in a batch. We find that mixing different graphs such as RNNLM and GNMT models with different number of layers results in both faster and better learning for RNNLM and GNMT with large number of layers (8-layer). As a matter of fact, both Placeto (Addanki et al., 2019) and HDP had problems matching human placement performance for 8-layer GNMT or 8-layer RNNLM. **With batch training, GDP is the first device placement work to match human expert performance for both 8-layer GNMT and 8-layer RNNLM. We also for the first time show that GDP-batch not only improves the search time (since we do not retrain the policy for every new graph), it can also improve the performance of the found placements.** More detailed results are shown in Appendix Table 5.

Table 2: Run time comparison on GDP-batch vs. GDP-one.

| Model | Speed up | Model | Speed up |
|---|---|---|---|
| 2-layer RNNLM | 0 | Inception | 0 |
| 4-layer RNNLM | 5% | AmoebaNet | -5% |
| 2-layer GNMT | 0 | 4-stack 36-layer WaveNet | 3.3 % |
| 4-layer GNMT | 0 | 2-stack 18-layer WaveNet | 15% |
| 2-layer Transformer-XL | 7.6% | 8-layer Transformer-XL | 1.5% |
| 4-layer Transformer-XL | 3% | | |

**Generalization to hold-out graphs**: Here we show another set of experiments where we treat GDP-batch as a pre-training strategy and remove the target graph from the batch training dataset. We then fine-tune the pre-trained model on the hold-out graphs for fewer than 50 steps, which takes less than one minute. We name this **GDP-generalization+finetune**. Figure 2 shows that GDP fine-tuning for hold-out graphs outperforms human expert placement and HDP consistently on all six batch training datasets, and performs only slightly worse than GDP-one. 2-layer RNNLM and 2-stack WaveNet almost match the performance of GDP-one. We also run inference (generate placement) directly on the pre-trained model for the target hold-out graphs, and name this **GDP-generalization-zeroshot**. We find that GDP-generalization-zeroshot only marginally hurts performance as compared to GDP-generalization+finetune, while being slightly better than human placement and HDP. This indicates that both graph embedding and the learned policies transfer and generalize to the unseen data.

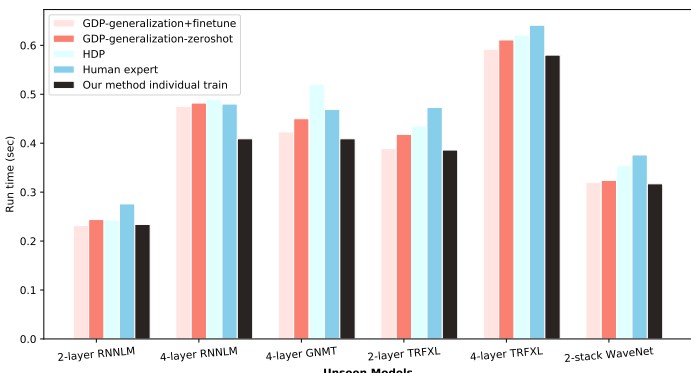

Figure 2: Finetuning on hold-out graphs.

**Comparisons with other generalized placement approaches**: Placeto (Addanki et al., 2019), to our knowledge, is the only other method beside GDP that shows true (and non-simulated) generalized device placement results. Direct comparison is not possible since Placeto uses a different hardware platform and different input graphs (Inception-V3, NMT, and NASNet). Placeto's search time is on average 2.65x faster than HDP, while GDP is on average 15x faster than HDP on our larger set of graphs. Apart from search time speed up, Placeto on average reduces placed graph run time by 3% (for its different graphs and hardware) while GDP on average reduces placed graph run time by 9.2%, compared to HDP. One advantage of GDP over Placeto is that it does not rely on any initial feasible placement. Providing a reasonable initial placement is often non-trivial for domain experts, especially for larger graphs such as 8-layer GNMT. As such, **we are the first to report superhuman results on 8-layer GNMT (with GDP-batch)**.

### 4.4 Ablation Studies

**Attention and Superposition.** We did an ablation study on the attention and the superposition layer in the transformer-XL placer network. We find that attention improves placement run time by an average of 18% compared to a placer network with no attention, and superposition improves placement run time by an average of 6.5% where all the graphs are trained in a single batch as described in Section 4.3. Without superposition network, batch training fails for AmoebaNet and Inception when mixing with larger RNNLM or GNMT models (4-layer).

**Pre-training graph embeddings.** We also evaluate a fine-tuning strategy by pre-training the graph embedding and placement network and fine-tuning the network on the down stream tasks. The difference here compared to Section 4.3 is that we also include the target graphs in the pre-training

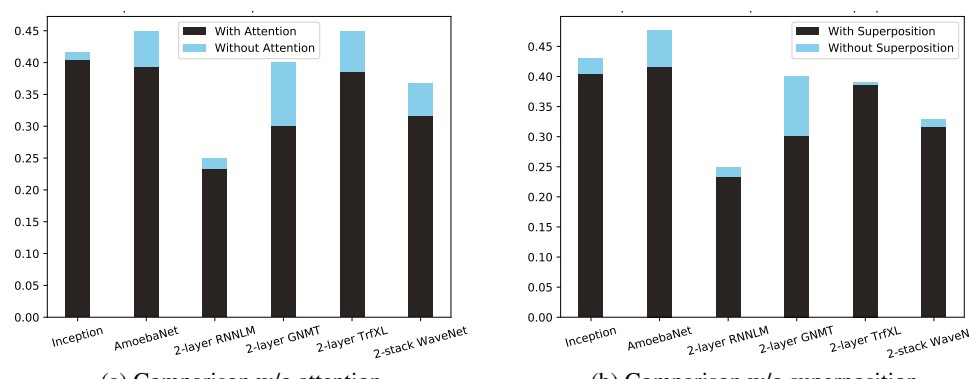

(a) Comparison w/o attention.       (b) Comparison w/o superposition.

Figure 3: Ablation Study on Attention and Superposition of the Placement Network.

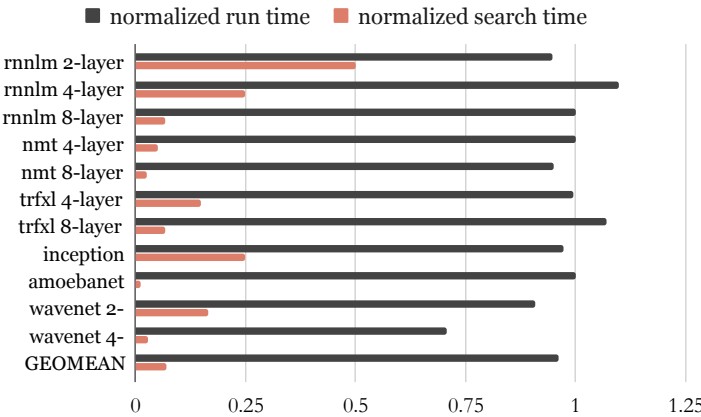

Figure 4: Normalized run time (step time for the generated placement) and normalized training time (search time) for fine-tuning. Time is normalized to GDP-one.

dataset. When GDP-batch is used as a pre-training strategy, the graph embedding and placement network assimilate meaningful graph representations and placement policies from a wide set of graphs, thus can be used as a strong baseline network for fine-tuning on downstream tasks. We compare the generated placement run time and the placement search time, normalized to GDP-one. We find that fine-tuning further reduces the the placed graph run time by an average of 5% and placement search time by an average of 86%, compared to GDP-one.

## 5 CONCLUSION

In this paper, we present a generalized device placement strategy that uses a graph neural network and super-positioning to generalize to arbitrary and held out graphs. Through experimental evaluation over a wide set of representative graphs from different domains including computer vision, speech, and NLP, we demonstrated over 15 times faster convergence while achieving a 16% and 9.2% reductions in step time over human expert placement and HDP, respectively.

## ACKNOWLEDGMENTS

TBD

## 6 Appendix

### 6.1 Proximal Policy Optimization

In device placement, the objective is to minimize the training step time of a given computational graph or a batch of dataflow graphs for a target system configuration (e.g. a 8-GPU cluster or a TPU pod), by placing operations onto different devices to enable model-level parallelism. This process corresponds to maximizing the expected performance in the MDP. For better sample efficiency, we adopted a Proximal Policy Optimization (PPO) (Schulman et al., 2017) algorithm. The objective is to maximize a surrogate objective:

$$L_\pi = E_{a_{[0:n]} \sim \pi} \big[ \frac{q'(a_n|s_n)}{q(a_n|s_n)} A_\pi(s_n, a_n) \big]$$

$$L_\pi = \max_{\pi'} \frac{1}{N} \sum_{n=0, a_n \sim \pi}^{N-1} \big[ \min(\frac{q'(a_n|s_n)}{q(a_n|s_n)}(R - \overline{R}), clip(\frac{q'(a_n|s_n)}{q(a_n|s_n)}, 1 - \epsilon, 1 + \epsilon)(R - \overline{R})) \big]$$

Within a loop, GDP PPO continuously samples placements from the distribution and evaluates their training times in real systems. For a rollout of $K$, we perform a minimatch of $m$ stochastic gradient ascent steps with respective to the objective of proximal policy optimization, which makes incremental policy improvements. The rollout steps $K$ and minibatch size $m$ are hyper parameters for PPO. We find a set of optimized hyper parameters and keep them fixed for all the experiments presented. As the rewards are generated on-the-fly based on real system measurements, we no longer need to re-evaluate the placement solutions in a separate phase.

### 6.2 Hyperparameters

In this section, we list out all the selected hyperparameters in our experiments for reproducibility in Table 3 and Table 4.

Table 3: Hyperparameters for Policy Network. $gs\_layers$: GraphSAGE layers, $gs\_knn$: GraphSAGE maximum neighbors, $trf\_d\_model$: Dimension of the TransformerXL model, $trf\_n\_head$: Number of attention heads, $trf\_layers$: Number of TransformerXL layers, $trf\_d\_heads$: Dimension of each attention head, $trf\_d\_inner$: Dimension of inner hidden size in positionwise feedforward.

| Parameters | Value | Parameters | Value |
|---|---|---|---|
| $gs\_layers$ | 4 | $gs\_dim$ | 128 |
| $gs\_knn$ | 5 | $trf\_layers$ | 2 |
| $trf\_d\_model$ | 128 | $trf\_n\_head$ | 5 |
| $trf\_d\_head$ | 25 | $trf\_d\_inner$ | 256 |

Table 4: Hyperparameters for PPO.

| Parameters | Value | Parameters | Value |
|---|---|---|---|
| $learing\ rate$ | 0.5 | $num\ of\ rollouts$ | 400 |
| $minibatches$ | 40 | $epochs$ | 5 |
| $epsilon$ | 0.2 | $entropy$ | 0.05 |

### 6.3 Input Graphs

We used a variety of widely used workloads from computer vision, speech, and NLP. In this section, we give a detailed explanation on the selected models and hyperparameters.

### 6.3.1 INCEPTION-V3

Inception-V3 (Szegedy et al., 2015) is a multi-branch convolutional network used for a variery of computer vision tasks, including classification, recognition, or generation. The network consists of blocks made of multiple branches of concolutional and pooling operations. Within a block, the branches of ops can be executed in parallel. However, the model is mostly sequential as the outputs of each block are concatenated together to form the input to the next block. We use a batch size of 64. The Tensorflow graph of this model contains 24,713 operations.

### 6.3.2 AMOEBANET

AmoebaNet (Real et al., 2018) is an automatically designed neural network that yields SoTA performance on ImageNet. Similar to Inception-V3, it contains Inception-like blocks called cells, which receives a direct input from the previous cell and a skip input from the cell before it. The network is made of redundant cells stacked together, therefore is more modular than Inception-V3. We use a batch size of 64. The Tensorflow graphs contains 9,430 operations.

### 6.3.3 RNNLM

Recurrent Neural Network Language Model (Zaremba et al., 2014; Jozefowicz et al., 2016) is made of many LSTM cells organized in a grid structure. The processing of each LSTM cell only depends on the results of 2 other cells (from the previous layer, and from the previous time step), which make the concurrent execution of many LSTM cells possible given enough hardware resources. We use batch size 64 and a hidden size of 2048. The corresponding TensorFlow graph contains 9,021 operations for a 2-layer model. The number of ops grow roughly proportional with the number of layers.

### 6.3.4 GNMT

Neural Machine Translation with attention mechanism (Bahdanau et al., 2015; Wu et al., 2016) has an architecture similar to that of RNNLM, but its many hidden states make it far more computationally expensive than RNNLM. To reduce the training time, prior work (Wu et al., 2016) propose placing each LSTM layer, as well as the attention and the softmax layer, on a separate device. This strategy demonstrates early success in human placement, we show that GDP can find significantly better placements. We use batch size 64. The original 2-layer encoder-decoder consisting of 28,044 operations. An extended 4-layer version consisting of 46,600 operations, An even larger 8-layer version consisting of 83,712 operations.

### 6.3.5 TRANSFORMER-XL

Transformer-XL (Dai et al., 2019) is an modified version of Transformer (Vaswani et al., 2017) that supports segement-level recurrence and a novel positional encoding scheme. This innovation enables learning dependency that is 80% longer than RNNs, and 450% longer than vanilla Transformers. We use a transformer-XL with batch size of 64, sequence length of 256, segment length of 64, model hidden dimension of 500 and feed forward hidden dimension of 1000, 10 heads, and head dimension of 50. The 2-layer Transformer-XL contains 2,618 operations. The number of ops grow roughly proportional with the number of layers.

### 6.3.6 WAVENET

WaveNet (van den Oord et al., 2016) is a generative model for speech synthesis. The model is fully probabilistic and autoregressive, with the predictive ditribution for each audio sample conditioned on all previous ones. Architecturally, WaveNet uses causal convolutions with dilations, to obtain a large receptive field. We use a WaveNet model with batch size 64 and a receptive field size of 2048 (9-layers per stack). An 5-stack WaveNet contains 4,374 operations and a 10-stack WaveNet contains 8,516 operations.

Table 5: Run time comparison on GDP batch training vs. the best of related methods (human expert, METIS, HDP, and GDP no batch training).

| Batch Setting | Model | speed up (s) |
|---|---|---|
| Batch 2 | Inception | 0 |
| | AmoebaNet | -4.5% |
| | 2-layer RNNLM | 0 |
| | 2-layer GNMT | 0 |
| | 2-layer Transformer-XL | 6.5% |
| | 2-stack 18-layer Wavenet | 4% |
| Batch 3 | 2-layer RNNLM | 0 |
| | 4-layer RNNLM | 0 |
| | 8-layer RNNLM | 4.5% |
| | 2-layer GNMT | 0 |
| | 4-layer GNMT | 0 |
| | 8-layer GNMT | 8% |
| Batch 4 | 3x8-layer GNMT | 5.1% |
| Batch 5 | 3x8-layer RNNLM | 4.5% |

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
