# OpenReview forum: "GDP: Generalized Device Placement for Dataflow Graphs"
_ICLR.cc/2020/Conference — Reject_

### Official Review · AnonReviewer3 · 2019-10-22
**Official Blind Review #3**

**Rating:** 3

**Review:**

GDP: Generalized Device Placement for Dataflow Graphs

This paper presents a method to assign the individual operations making up the dataflow graph of a deep neural network to a set of connected devices on which they should be executed, with the objective of maximizing runtime performance of inference. The paper’s proposed approach relies on graph neural networks to produce embeddings of the dataflow graph nodes that are claimed to be transferable to unseen graphs. These embeddings serve as input to a transformer-based sequence model that outputs the final assignment. The end-to-end model is trained with a reinforcement learning criterion minimizing the expected placement runtime. Experimental comparisons against alternative placement methods (including human expert) are presented for several large neural networks of different architectures.

Overall, the paper is well written, easy to follow, and addresses an important concern is scaling up neural networks to large model sizes. The proposed approach combining graph neural networks with transformer-based placement network appears, at first glance, novel.

My main reservations with the paper is that it lacks many details in several key sections, preventing a full appreciation of the contributions, and making reproducibility of results impossible to contemplate. In particular:

1. How the placement network is designed and used (section 3.2) is completely lacking.
2. The details of how the training and testing datasets are obtained are also lacking. In particular, the specific model architectures for the likes of RNNLM, including hyperparameter choices, the runtime data fed to those models, etc. (This can be given in appendix). It is also not clear if the models listed in Table 1 are singular models with fixed values of hyperparameters, or if they correspond to distributions over models (e.g. with differences in hyperparameters).
3. How PPO is used is glossed over: for instance, is the proposed placement of a given model tried « live » in the inner loop of PPO to generate the reward corresponding to its runtime? What runtime data is fed to the model in order to do this? Since this would be a somewhat unusual setup, an illustration of the overall training loop would certainly solidify understanding. Moreover, a few sentences or equations explaining how PPO is used in this context would also help. Is training with PPO imperative for getting good model performance? Have other methods been tried?

In section 3.3, the parameter superposition mechanism appears to perform a kind of meta-learning. Explicit connections to such should be made, such as « Tadam: Task dependent adaptive metric for improved few-shot learning » (NeurIPS 2018).

Moreover, at the end of section 4.4, the paper claims to « report superhuman results on 8-layer GNMT ». Unless this reviewer misunderstands, the results in Table 1 (wherein GDP-one clocks in at 0.649 for 8-layer GNMT, versus 0.610 for Human Placement) would contradict this claim. This should be clarified.

Given these reservations, in spite of the potential of the proposed approach, the paper appears in its current form too immature to recommend acceptance at ICLR.


**Experience Assessment:**

I do not know much about this area.

**Review Assessment: Checking Correctness Of Derivations And Theory:**

N/A

**Review Assessment: Checking Correctness Of Experiments:**

I assessed the sensibility of the experiments.

**Review Assessment: Thoroughness In Paper Reading:**

I read the paper at least twice and used my best judgement in assessing the paper.

---

> ### Author Response · Authors · 2019-11-13
> **Response to Review #3: We added more details.**
>
> Please find our PDF rebuttal: https://drive.google.com/file/d/1xosZB6qkoCyFr-LvuNpX_eybfWVZZp4J/view
>
> We greatly appreciate the reviewer's feedback on clarifying technical details and design choices. For most the questions raised by the reviewer, we have updated our paper to provide better design details on the controller network in Section 3 and Section 3.2 and more in-depth explanation on the usage of PPO and hyperparameter selections. We find those changes have improved the paper significantly.
>
> Q1: "How the placement network is designed and used (section 3.2) is completely lacking.":
> We added Section 3 and Section 3.2 in the updated paper on the design philosophy and details. Please find the related updated sections in the paper.
>
> Q2: "The details of how the training and testing datasets are obtained are also lacking.":
> We added an Section 6.3 in the Appendix to explain the details of our dataset. Our pre-training datasets consists of various models from six widely used workloads as described in Appendix Section 6.3. We vary the number of layers of these six workloads and create our training datasets from these models. For the results presented for GDP-batch in Table 2 in Section 4.3, we mix 11 graphs to create a batch of data for training, and found batch training improved learning the placement for individual graphs. For the generalization results in Figure 2 in Section 4.3, we have a pre-training and test phase, where the test can be based on finetuning or zero-shot. The pre-training data contains a mixture of all six workloads with varied number of layers except the test workload. For example, for the test of 2-layer RNNLM, we exclude all RNNLM models and use a training set consisting of Inception, AmoebaNet, GNMT, Transformer-XL, and WaveNet, with variable number of layers (2-layer, 4-layer, and  8-layer). We will add the details in the final version.
>
> Q3: "How PPO is used is glossed over:..Have other methods been tried?"
> We added an Appendix in the updated paper. Section 6.1 discusses detailed formulation and usage of PPO and Section 6.2 listed our hyperparameters across all the experiments.
> In device placement, the objective is to minimize the training step time of a given computational graph or a batch of dataflow graphs for a target system configuration (e.g. a 8-GPU cluster or a TPU pod), by placing operations onto different devices to enable model-level parallelism. This process corresponds to maximizing the expected performance in the MDP. For better sample efficiency, we adopted a Proximal Policy Optimization (PPO) [1] algorithm.
> The objective is to maximize a surrogate objective:
> \begin{equation*}
>     L_{\pi}=E_{a_{[0:n]}\sim\pi}[\dfrac{q_{'}(a_{n}|s_{n})}{q(a_{n}|s_{n})}A_{\pi}(s_{n},a_{n})]
> \end{equation*}
> \begin{equation*}
>     L_{\pi}=\max\limits_{\pi^{'}} \dfrac{1}{N}\sum_{n=0,a_{n}\sim\pi}^{N-1}
>     [\min (\dfrac{q_{'}(a_{n}|s_{n})}{q(a_{n}|s_{n})}(R -\overline{R}), clip(\dfrac{q_{'}(a_{n}|s_{n})}{q(a_{n}|s_{n})}, 1-	\epsilon, 1+\epsilon)(R -\overline{R}))]
> \end{equation*}
> \begin{equation*}
> \end{equation*}
>
> Within a loop, GDP PPO continuously samples placements from the distribution and evaluates their training times in real systems. For a rollout of $K$, we perform a minibatch of $m$ stochastic gradient ascent steps with respective to the objective of proximal policy optimization, which makes incremental policy improvements. The rollout steps $K$ and minibatch size $m$ are hyper-parameters for PPO. We find a set of optimized hyper parameters and keep them fixed for all the experiments presented. As the rewards are generated on-the-fly based on real system measurements, we no longer need to re-evaluate the placement solutions in a separate phase.
>
> We originally used Policy Gradient, but later found PPO yields better convergence time and sample efficiency. For our problem particularly, sample efficiency and training time are very critical.
>
> Q4: "Explicit connections to such should be made":
> We thank the reviewer for pointing us to this work. We add this citation in our updated version. Indeed the proposed method of coupling three identified elements (metric scaling, task conditioning, auxiliary task co-training) in few-shot learning is very relevant to the problem we are tackling and relevant to our proposed solutions.
>
> Q5: "The paper claims to « report superhuman results on 8-layer GNMT ».":
> The superhuman number was achieved by batch training (GDP-batch), as explained in Section 4.3 (the last paragraph). The 0.649 second was achieved using GDP-one, which performs slightly worse than GDP-batch in this case.
>
> [1] Proximal Policy Optimization Algorithms

---

> > ### Author Response · Authors · 2019-11-13
> > **Response to Review #3: More details on Q1**
> >
> > More particularly, we added below paragraphs:
> >
> > Section 3: Our proposed policy network $\pi_\theta$ consists a graph embedding network that learns the graphical representation of any dataflow graph, and a placement network that learns a placement strategy over the given graph embeddings. The two components are jointly trained in an end-to-end fashion. The policy $p(a|G)$ is applied to make a set of decisions at each node. These decisions, denoted as $a_{v}$ for each $v \in V$ across all nodes, form one action $a=\{a_{v \in V}\}$. One decision corresponds to playing one arm of a multi-bandit problem, and specifying the entire $a$ corresponds to playing several arms together in a single shot. Note the architecture is designed to be invariant over the underlying graph topology, enabling us to apply the same learned policy to a wide set of input graphs with different structures.
> >
> > Section 3.2: The graph neural network works as a feature aggregation network that learns a trainable feature representation for the computational graph, we still need a policy network that produces actions on a per node basis. Given $h_{v}$'s, the policy network produces $a_{v}$'s through conditionally independent predictions, where the prediction for one node $v$ does not depend on the prediction of other nodes.
> >     \begin{align}
> >         p(a|G)=\prod_{v}p(a_{v}|G)=\prod_{v}p(a_{v}|f(h_{v}))
> >     \end{align}
> > While $f$ can be represented using multilayer perceptrons (MLPs), where the MLPs are shared across all nodes for prediction the placement output distributions. However, MLPs lack a dependency tracking mechanism across nodes. In practice, the placement of one node can be determined by the placement of another node, where the placed node may consume a large size of data produced by the other node. Intuitively, an attention network can learn this dependency and the relative importance of dependencies across an entire graph. Therefore, we decided to use an attention-based placement network to better track inter-node placement-related dependencies.

---

### Official Review · AnonReviewer2 · 2019-10-24
**Official Blind Review #2**

**Rating:** 6

**Review:**

In this paper the authors propose an end-to-end policy for graph placement and partitioning of computational graphs produced "under-the-hood" by platforms like Tensorflow. As the sizes of the neural networks increase, using distributed deep learning is becoming more and more necessary. Primitives like the one suggested by the authors are very important in many ways, including improving the ability of the NN to process more data, reduce energy consumption etc. The authors compared to prior work propose a method that can take as input more than one data flow graphs, and learns a policy for graph partitioning/placement of the operations in a set of machines that minimizes the makespan. This problem in principle is NP-hard as it entails both graph partitioning and graph scheduling as its components. The authors propose a heuristic that composes of two existing methods: graph neural networks are used to produce an embedding of the computation/data flow graph, and then a seq-2-seq placement network. The method is able to generalize to unseen instances.

I vote for weak reject since some issues that I would like to see addressed by the author(s) are not. These include the fact that since the goal is to minimize the makespan,  scheduling within each machine the operations should be addressed in a better way. Also, while the objective J(\theta) is reasonable, the distribution for the makespans could be very skewed (e.g., heavy tails over the dataflow graphs). Doesn't this affect the results? Finally, the novelty from a deep learning perspective is limited.

- How do the authors address the issue of scheduling the operations within each machine?
- How is \mathcal{D} formally defined (i.e., the range of the mapping function)? Do you take into account the different number of machines, their memory footprints that can be significantly different, the different processing units they may have (GPUs, CPUs, TPUs)? Is the number of machines used for the partition automatically learned by the policy? That part was not very clear.
- Since the authors compare with METIS, it is worth also comparing with Scotch https://www.labri.fr/perso/pelegrin/scotch/ that is also publicly available.
- Can the authors comment on the scalability of their method as a function of n (number of nodes), and k (number of  devices)?


Update: Thanks to the author(s) for the detailed feedback. I have upgraded my score accordingly.

**Experience Assessment:**

I have published one or two papers in this area.

**Review Assessment: Checking Correctness Of Derivations And Theory:**

I carefully checked the derivations and theory.

**Review Assessment: Checking Correctness Of Experiments:**

I carefully checked the experiments.

**Review Assessment: Thoroughness In Paper Reading:**

I read the paper thoroughly.

---

> ### Author Response · Authors · 2019-11-13
> **Response to Review #2:  We thank the reviewer's insightful feedback.**
>
> PDF link:  https://drive.google.com/file/d/1xosZB6qkoCyFr-LvuNpX_eybfWVZZp4J/view
>
>
> Q1: "..scheduling within each machine the operations should be addressed in a better way":
> We acknowledge that scheduling is indeed an important consideration in minimizing the step time (i.e., makespan) for a graph. However, in this paper we consider learning to schedule as an orthogonal problem. The focus of this paper is on partitioning the graph through op placement to minimize step times as well as satisfy memory constraints of devices. Our key contributions are towards addressing some of the shortcomings of prior work (e.g., lack of generalization, efficiency of search) that end up being significant bottlenecks for practical adoption. We believe that the ideas introduced in this paper are novel and significant on their own. However, we are actively working on learned approaches to scheduling. Below we discuss some of our preliminary work in addressing scheduling and outstanding challenges.
>
> Current approach: First, for the current version of the paper, we do indeed rely on the default scheduling strategy for TensorFlow. The default strategy is to maintain a per-device ready queue of operations and execute ops in a first-come-first-serve manner. We have incorporated this detail in our paper.
>
> Learning a better schedule: We are currently investigating alternate scheduling strategies based on priorities. We substituted the FIFO scheduler with a priority scheduler that dequeues ops from the ready queue in order of priority.  More details are discussed in the PDF file. Our preliminary results are promising and demonstrate around 10% further reduction in step times using both of these approaches. However, these results were obtained using a simulator. As mentioned in response #1 for Reviewer 1, based on our experience it is critical to ensure reproducability of simulation results on an actual device. However, this currently has significant engineering challenges due to a lack of easy API to plug into a priority based scheduler. We are actively working on overcoming these challenges. Other works like REGAL [3] that address scheduling in addition to placement present simulation-only results.
>
> Q2: "...the distribution for the makespans could be very skewed":
> We assume that the reviewer's concern is how the potentially large variation in step times (i.e., makespans) for the graphs in our workload is handled.  We applied two techniques to handle the skewness in data distribution:
>
> 1. The superposition network layer as described in Section 3.3 orchestrates the training of parameters with different input graphs. This largely mitigates the interference created by different input graphs during training. Consequently, we do not favor updates from input graphs that are easy to optimize for a bigger run time improvement. Our empirical results in Section 4.3 also confirms that batch training for multiple graphs with different distributions won't negatively affect individual graph placement run time. Our ablation study in Section 4.4 also confirms that the superposition layer contributed 6.5% of average run time improvement for GDP-batch.
>
> 2. The reward is computed using the negative square root of normalized run time, where the run time is normalized with the best run time achieved in baseline models (e.g. the best of human placement, METIS, and HDP.). This ensure that the RL does not prioritize updates that benefit graphs that take long to run vs. graphs that don't take long to run.
>
> Q3: "...novelty from a deep learning perspective is limited":
> Please see our response to Q1 from Reviewer 1.
> We do think that innovative application of pre-existing deep learning techniques to new domains and problems is a significant enough contribution and of sufficient interest to ICLR audience. In fact, ICLR has accepted such papers [1] in the recent past.
>
> Q4: "How is $\mathcal{D}$ formally defined..."?
> $\mathcal{D}$ is the search space of our placement, which is one CPU and n GPU nodes. In our experiments, we set n=2,4,8, which is a standard approach used by related work (e.g. HDP [1]). Since our rewards are based on real systems measurement, we base our search space on realistic GPU cluster configuration and do not invent arbitrary new configurations (e.g. a mixture of GPUs and TPUs).
>
> Q5: "...worth also comparing with Scotch":
> As presented in HDP [1], Scotch is much worse than human placement, while human placement is much worse than HDP. Our method beat HDP in all aspects including graph run time (10%) and search time (15x reduction).
> We will be happy to incorporate a direct comparison to Scotch in the final version of the paper.
>
> [1] Hierarchical Device Placement, ICLR 2018.
> [2] Placeto: Learning Generalizable Device Placement Algorithms for Distributed Machine Learning, Arxiv.
> [3] Reinforced Genetic Algorithm Learning for Optimizing Computation Graphs, in submission to ICLR 2020.

---

> > ### Author Response · Authors · 2019-11-13
> > **Response to Review #2: One more question addressed here.**
> >
> > Q6: "Can the authors comment on the  scalability of their method as a function of n (number of nodes), and k (number of  devices)?":
> > Our controller model parameters are invariant to number of nodes in the graph, therefore parameter-wise GDP is insensitive to the number of nodes in the graph. Computationally, we only apply a modified Transformer-XL encoder in our placement network. The parallel attention network runs on modern GPU efficiently, which is much faster than a conventional progressive RNN-based solution (e.g. the controller network used by HDP[1] and Placeto [2]). We also empirically find that the training time scales linearly with the number of nodes n, while the number of devices k has little impact on the training time of our controller.
> >
> > [1] Hierarchical Device Placement, ICLR 2018.
> > [2] Placeto: Learning Generalizable Device Placement Algorithms for Distributed Machine Learning, Arxiv.
> > [3] Reinforced Genetic Algorithm Learning for Optimizing Computation Graphs, in submission to ICLR 2020.

---

### Official Review · AnonReviewer1 · 2019-10-25
**Official Blind Review #1**

**Rating:** 6

**Review:**

Summary: This work proposes to use a combination of graph neural networks (GNNs) and proximal policy optimization (PPO) to train policies for generalized device placement in dataflow graphs. Essentially, (1) a GNN is used to learn representations of a dataflow graph (in an inductive manner), (2) a transformer is used to output a device placement action for each node in the graph, and (3) the entire system is trained end-to-end via PPO. Extensive experimental results show very impressive results compared to strong baselines.

Assessment: Overall, this is a solid application paper. The authors GNNs, PPO, and Transformers in an effective, well-motivated, and sound manner. Moreover, the task is interesting and relevant. There is not significant methodological novelty, as the authors are essentially combining standard components in a straightforward way. That said, the results are strong and the paper is well-written, so it certainly has merits as an application paper.

Will code be released? This is essential for reproducibility, as the paper does not contain sufficient technical details to allow for reproduction.

Reasons to accept:
- Strong empirical results on an interesting application
- Well-written paper
- Thorough experiments

Reasons to reject:
- Incremental methodological contribution
- Likely difficult to reproduce

**Experience Assessment:**

I have published in this field for several years.

**Review Assessment: Checking Correctness Of Derivations And Theory:**

I assessed the sensibility of the derivations and theory.

**Review Assessment: Checking Correctness Of Experiments:**

I assessed the sensibility of the experiments.

**Review Assessment: Thoroughness In Paper Reading:**

I read the paper at least twice and used my best judgement in assessing the paper.

---

> ### Author Response · Authors · 2019-11-13
> **Response to Review #1: PDF version in https://drive.google.com/file/d/1xosZB6qkoCyFr-LvuNpX_eybfWVZZp4J/view.**
>
> We thank the reviewer for acknowledging that our work is "a solid application paper" and "the results are strong and the paper is well-written". Indeed, we agree that the key contribution of our paper is combining techniques from existing work to solve the device placement problem.
>
> Q1. "Incremental methodological contribution":
> While the final solution presented in the paper may seem straightforward, arriving at this design required a deeper understanding of the underlying system (i.e., the compiler and the execution engine) as well as exploration and experimentation over a number of alternatives. To highlight the complexity of the process, we discuss some of the key considerations that finally led us to this seemingly straightforward solution.
>
> 1. Focus on workloads where model parallelism is truly relevant -- The computational capability and available high-bandwidth memory for modern accelerators is often sufficient to handle most machine learning models with a few thousand nodes and moderately large number of parameters. For such models, the best placement is usually the degenerate placement of ops on a single device coupled with data parallelism. One of the key distinguishing factors of our work over other related works (e.g. Placeto [3] and REGAL [4]) is our focus on substantially more complex model architectures which have tens of thousands of nodes in their dataflow graph. While there may be few such models, they are often widely used for many important applications. We believe it is these models where device placement truly becomes relevant and necessary. Finding placements for graphs with such large number of nodes led us to design a network that (a) can make placement decisions for the entire graph in a single shot, not like the progressive solutions proposed in Hierarchical Device Placement [2] and Placeto [3] and (b) uses a re-engineered transformer-XL (Section 3.2) that is able to learn dependencies even over really long ranges (for instance, over graphs with 50k nodes).
>
> 2. Generalization as major hurdle for integration with the compiler -- Compilers for ML graphs often have stringent latency requirements. Prior approaches [1][2] learn a policy for placing ops that is specific to a single graph. Given that learning the policy itself is slow and compute intensive, it is infeasible to learn the policy on the fly at compile time. A key contribution of our work is learn a general policy and demonstrate that it can achieve SOTA results even over a heterogeneous set of large graphs. To accomplish this, we rely on (a) a graph-embedding network that encodes operation features and dependencies into a trainable graph representation and (b) use super-positioning based on the input graph embeddings to effectively orchestrate the optimization dynamics of graphs with drastically different sizes and characteristics.
>
> 3. Simulation versus true device measurements -- While we did have access to a well-designed simulator, we found that simulators make simplifying assumptions that often lead to discrepancy between simulation measurements and measurements on the actual device. For instance, simulators typically work well on only well known benchmarks but not on unexpected placements or on arbitrary graphs. Secondly, simulators often omit certain optimizations (like swapping of tensors to-from device memory) that are more dynamic in nature and are therefore hard to simulate statically. While using simulation enables faster convergence, the generated placements are often fragile. Therefore, we explicitly decided to evaluate the proposed placements on a true device even during training. However, given that true measurements are expensive, we resorted to using more sample-efficient learning algorithms like PPO instead of Policy Gradients to reduce the search time substantially.
>
> Q2. "Likely difficult to reproduce":
> We acknowledge that the original draft of the paper lacked some of the implementation details that could have made it difficult to reproduce the results in the paper. To improve reproducability, we have added more technical details in the revised version. Specifically, we included a better explanation of the controller network in Section 3, added Appendix 6.1 discussing the details of PPO algorithm and parameters, added Appendix 6.2 with details about the hyperparameters used, and finally added Appendix 6.3 containing a thorough explanation of the input graphs. We have been working on an open source solution (including a simulation tool to enable placement over non-existent device configurations) and plan to release the code upon the paper's acceptance.
>
> [1] Device Placement Optimization with Reinforcement Learning, ICML 2017.
> [2] Hierarchical Device Placement, ICLR 2018.
> [3] Placeto: Learning Generalizable Device Placement Algorithms for Distributed Machine Learning, Arxiv.
> [4] Reinforced Genetic Algorithm Learning for Optimizing Computation Graphs, in submission to ICLR 2020.

---

### Public Comment · ~Peng_Wang2 · 2019-11-08
**Like the work.**

We are finding scheduling and placement of large graph is important for application. Read this work, and I think the proposed strategy is in principle simple to reproduce (Just PPO + Graph Ebedding with SAGE), Personally, I think the comments from R#2 is a bit critical since placement and scheduling are orthoganal directions to optimize, and the author has already shown superior performance with placement only. Simutaneously do scheduling is a plus, but not the reason for rejection.

In addition, since this paper is majorly about successfully applying existing deep learning strategies to system optimization,  therefore, domain specific engineering and combination of domain knowledge is the novel part. I think it is unfair to look at the work with a deep learning perspective.

---

### Author Response · Authors · 2019-11-13
**General Reponse to All Reviewers**

We thank all reviewers for the hard work and insightful suggestions. We have updated a version of our paper, and provided clarification in the following:
1. Design rationales and details on the placement network in Section 3 and Section 3.2.
2. An appendix with more explanation on the PPO algorithm (how we do rollouts and policy updates), hyperparameters for the controller network and PPO algorithm, and input graphs (neural architectures and hyperparameters).

We believe that our paper is novel on its own and sheds light on applying ML to real systems design and landing of AI, as we explained in Response to Q1 from Rev1.

---

### Decision · Program_Chairs · 2019-12-19

**Decision:**

Reject

**Comment:**

This paper presents a new reinforcement learning based approach to device placement for operations in computational graphs and demonstrates improvements for large scale standard models.

The paper is borderline with all reviewers appreciating the paper even the reviewer with the lowest score. The reviewer with the lowest score is basing the score on minor reservation regarding lack of detail in explaining the experiments.

Based upon the average score rejection is recommended. The reviewers' comments can help improve the paper and it is definitely recommended to submit it to the next conference.